# Origin of the Photoluminescence of Metal Nanoclusters: From Metal-Centered Emission to Ligand-Centered Emission

**DOI:** 10.3390/nano10020261

**Published:** 2020-02-04

**Authors:** Tai-Qun Yang, Bo Peng, Bing-Qian Shan, Yu-Xin Zong, Jin-Gang Jiang, Peng Wu, Kun Zhang

**Affiliations:** Shanghai Key Laboratory of Green Chemistry and Chemical Processes, College of Chemistry and Molecular Engineering, East China Normal University, Shanghai 200062, China; tqyang@chem.ecnu.edu.cn (T.-Q.Y.); 51174300109@stu.ecnu.edu.cn (B.P.); 52184300051@stu.ecnu.edu.cn (B.-Q.S.); 51174300121@stu.ecnu.edu.cn (Y.-X.Z.); jgjiang@chem.ecnu.edu.cn (J.-G.J.)

**Keywords:** photoluminescence mechanism, metal nanoclusters, quantum confinement effect, ligand effect, p band intermediate state (PBIS), interface state, nanocatalysis

## Abstract

Recently, metal nanoclusters (MNCs) emerged as a new class of luminescent materials and have attracted tremendous interest in the area of luminescence-related applications due to their excellent luminous properties (good photostability, large Stokes shift) and inherent good biocompatibility. However, the origin of photoluminescence (PL) of MNCs is still not fully understood, which has limited their practical application. In this mini-review, focusing on the origin of the photoemission emission of MNCs, we simply review the evolution of luminescent mechanism models of MNCs, from the pure metal-centered quantum confinement mechanics to ligand-centered p band intermediate state (PBIS) model via a transitional ligand-to-metal charge transfer (LMCT or LMMCT) mechanism as a compromise model.

## 1. Introduction

Metal nanoclusters (Au, Ag, Pt, Cu) are a new class of attractive materials owing to their enhanced quantum confinement effect, which endows them with unusual optical and electronic properties [1,2,3,4,5,6]. Au and Ag NCs in particular have attracted tremendous interest because of their wide applications in single-molecule studies, sensing, biolabeling, catalysis, and biological fluorescence imaging [7,8,9,10,11,12,13,14,15,16]. The physical and chemical properties of metal nanoclusters (MNCs) are highly dependent on their size, shape, composition, and even assembly architecture [17,18,19,20]. These clusters are composed of a few to hundreds of atoms with a size approaching the Fermi wavelength of an electron (~0.5 nm for Au and Ag), and they possess discrete molecular-like electronic energy band structures, resulting in intense light absorption and emission. Luminescent MNCs exhibit outstanding optical properties including a large Stokes shift (generally large than 100 nm), large two-photon absorption cross-section, good photostability, good biocompatibility, and the emission wavelength could be easily adjusted by controlling their size and surface ligands, which are not present in conventional organic dyes [1,9,11,21]. Their excellent luminous performance is attractive for the applications in biomedicine, since it provides avenues for designing optical sensors, biolabeling, bioimaging, photosensitizers, and light-emitting devices. However, the luminous quantum yield of MNCs is still not competitive when compared to those organic dyes and semiconductor quantum dots (QDs), which always hinder their practical application. One primary reason is the lack of full understanding of the luminescence origin of MNCs at the molecular level, since its photoluminescence properties depend on many factors including cluster size, surface ligands architecture, cluster assembly structure, etc. The preparation and application of MNCs have been previously reviewed [2,8,10,14,15,22,23,24,25]. Herein, we are mainly focused on the photoluminescence emission origin of MNCs. It is imperative to clarify the origin of photoemission of MNCs and thus improve their luminous efficacy. Two major explanations about the photoluminescence mechanism have become gradually accepted by researchers. One is the pure metal quantum confinement effect [1,26,27,28]. That is, as the size of MNCs approaches the Fermi wavelength of metals (usually <1 nm), the continuous band of energy level becomes discrete, leading to the emerging of molecule-like properties. The emission of MNCs was originated from intraband (sp–sp) and interband (sp–d) transitions [29,30]. The other explanation is the charge transfer on the shell of metal clusters due to the interaction between the functional ligands and the metal core, i.e., ligand-to-metal charge transfer (LMCT) [31] or ligand-to-metal−metal charge transfer (LMMCT: emphasize the gold–gold interactions resulted from the relativistic effect) [32] mechanism borrowing from the concept of PL emission of the transition organometallic complexes. Both of these suggest that the intraband (sp–sp) and interband (sp–d) transitions from metal NCs probably act an intermediate state (or dark state) to determine the electron transfer [33]. These two explanations are proposed to explain the size-dependent and ligand-dependent emission phenomenon respectively, but they are self-contradicted to illustrate the size-dependent and size-independent emission. In addition, more and more studies show that nonluminescent groups could serve as chromophores in a certain solvent condition or confined nanospaces. In addition, more and more strong evidences from recent studies tend to prove the limitation of metal-centered LMCT and LMMCT models to explain the photoluminescence (PL) properties of metal NCs.

So far, two main approaches—the bottom–up and top–down strategies—have been developed for the synthesis of emission wavelength adjustable luminescent MNCs with high quantum yield (QY) in large quantities, as illustrated in Figure 1 [8,9,10,14,15,34]. The size effect (including metal corearchitecture) [35,36,37,38,39,40], ligand effect (charge-donating capability [33], chirality [41,42], hydrophily [43]), valance state of surface metals [33,44], and superlattice structures of assembly of metal clusters [45,46] have been intensively studied.

It is important to note that especially for the bottom–up synthetic strategy, a common feature of luminescence metal NCs is that the templates used for the synthesis as protective ligands, such as nonconjugated polymers, proteins, amino acid molecules, and even small chemical molecules, always contain electron-rich heteroatoms, including oxygen (O), nitrogen (N), sulfur (S), phosphorus (P), and others, emphasizing the dominant role of the surface ligand to tune the PL properties. However, the exact role of the surface ligand and the dynamic and/or kinetic of electron relax at the nanoscale interface between core–shell structured MNCs was not clarified. In this report, we simply review the present fundamental understanding of the photoluminescence origin of MNCs, especially our recently developed the ligand-centered p band intermediate state (PBIS) model to elucidate the abnormal PL emission phenomena. Finally, we proposed some forward-looking perspectives to the future developments of MNCs, in particular, for the application of MNCs in the nanocatalysis science.

## 2. Photoemission Mechanism of Metal Nanoclusters

### 2.1. Size-Dependent Photoemission

As the saying goes, “gold will glitter forever!” The first experimental observation of photoluminescence from bulk gold metal can date back to as early as 1969 [29]. Mooradian reported that bulk gold and copper film could generate photoemission when excited under a high-energy laser. The emission was arising from the interband transitions between electrons in conduction-band states below the Fermi level and holes in the d bands generated by optical excitation, as shown in Figure 2. This is the first proposed energy band structure to explain the photoemission mechanism of transition bulk metal. The energy gap was elucidated to 2.0 eV for copper and 2.2 eV for gold with the help of emission spectra. While photoemission was observed from bulk metal, the emission was generated under an extreme condition, and the quantum yield was very low (10^−10^), which was not suitable for practical application. Metal nanoparticles with a size range from 2 to 50 nm exhibit intense colors due to the surface plasmon resonance (SPR); in some cases, photoluminescence was also observed in these small metal nanoparticles, which contrasts the conclusion that MNCs with SPR effect generally do not produce fluorescence [2]. In 1998, Wilcoxon et al. observed relatively intense photoluminescence when the size of the metal nanocluster was sufficiently small [47]. Nonluminescent gold particles can also become luminescent by partial etching with potassium cyanide (KCN). The photoemission with large Stokes shift is ascribed to the electron and hole interband recombination process. When the particle size was further reduced to 2 nm or less, the continuous band structure will be broken down into discrete energy levels according to quantum confinement mechanics, exhibiting molecular-like properties and not exhibiting typical plasmonic properties. Indeed, these very small transition metal clusters with precisely defined architectures are first studied in heterogeneous catalysis [48], which did not attract much more attention from photochemists and photophysicists. With the rapid development of synthetic nanochemistry, more and more people focused on the peculiar optoelectronic properties of these tiny metal clusters, even for still low quantum yield (QY) confined in a gas matrix at low temperature [49,50]. By combining solid-state principles as well as a molecular model, Link et al. improved the energy band structure model to explain the electron transition process and subsequent PL emission, which was attributed to the intraband (sp–sp) and interband (sp–d) transitions as illustrated in Figure 3 [30].

With the progress of wet chemical synthesis strategy, luminescent water-soluble metal nanoclusters with different sizes were successfully synthesized with improved PL emission efficiency and tunable colors [26,51,52,53,54]. The photoemission wavelength could be varied from the ultraviolet (UV) to near-infrared (NIR) region by controlling the sizes of metal NCs [1]. In the early 2000s, Dickson et al. first achieved the preparation of water-soluble Au NCs with high fluorescence in solution using a biocompatible dendrimer (OH-terminated poly(amidoamine) (PAMAM)) as a capping agent [26,51,52]. The emission wavelength of as-prepared Au NCs could be adjusted from UV to the NIR region by changing the metal/polymer ratio during the synthesis. Benefiting from the progress of soft-ionization mass spectrometry (MS) technology, the cluster size could be precisely identified. By ingeniously creating the linear correlation between the emission intensity at a certain wavelength and the intensity of mass spectra at a certain cluster size, the emission wavelength of different clusters was determined. The correlation between the size of Au nanodots and emission energies was established and fit a simple scaling relation of *E*_Fermi_*/N*^1/3^ (where *E*_Fermi_ is the Fermi energy of bulk gold and *N* is the number of Au atoms) [26], as shown in Figure 4. The size-dependent scaling of excitation and emission energies with *E*_Fermi_*/N*^1/3^ confirms the good matching with the jellium model where the gap between the discrete 5d valence band and the 6sp conduction band decreases with the increasing cluster size [1]. If this model is rational, the coinage metals at the same family with the similar size or the close atom numbers should show very similar PL emission. However, the size distribution and optical properties of stable “magic” number Ag NCs were found to be obviously different to the gold NCs. Recently, Udaya Bhaskara Rao et al. successfully synthesized Ag NCs with very similar particle size (Ag_7_ and Ag_8_) using mercaptosuccinic acid (H_2_MSA) as capping ligands; they showed distinguished PL emission at approximately 440 nm and 650 nm for Ag_7_ and Ag_8_ NCs, respectively. Only one silver atom difference in cluster structures could lead to such a big change of emission color from blue to red [55]. This could not be simply exchanged only by a size-dependent quantum confinement effect (QCE).

Even though MNCs shows strong size-dependent PL emission properties, the quantum lifetime and QY calculated by theoretical calculation are much less than the experimental observed values, and if we are only considering the contribution of metal to PL emission, the wavelength of PL emissions calculated by the theoretical model should lie in a near-infrared range, instead of the observed visible wavelength emission. These disagreements between experimental and theoretical data indicate the invalidity of metal-centered quantum confinement mechanics. 

### 2.2. Size-Independent Photoemission 

With sophisticated synthetic techniques, atom-precise MNCs were achieved for a number of metal nanoclusters (referred to as M_n_L_m_, where n and m are the numbers of metal atoms and ligands in the cluster). Very recently, Luo and his co-workers synthesized a series of glutathione (GSH) protected Au NCs with precisely atomic compositions (Au_29_SG_27_, Au_30_SG_28_, Au_36_SG_32_, Au_39_SG_35_, Au_43_SG_37_), and, to one’s surprise, they all exhibited completely the same emission at approximately 610 nm [56], which is completely contradictory to the explanation of the classical QCE mechanism. Furthermore, the emission wavelength could be readily regulated from 600 nm to 810 nm by just fine-tuning the surface ligands’ coverage [57], suggesting the pivotal role of surface-protective ligands to tune the PL of Au NCs. In addition, the Au NCs with the same number of core atoms but different protecting ligands show different PL properties [33,58,59]. These results clearly indicate that the metal core is not the only determining factor for the photoemission of MNCs. Other components of MNCs including the nature of coordinate ligands, valence states of surface metal atoms, and assembly architectures of nanoclusters also make a significant difference on the emission properties of MNCs.

#### 2.2.1. Ligand Effect

It is well accepted that the organic/inorganic scaffolds and/or surface protective ligands were used to stabilize the metal nanoclusters, since naked MNCs would strongly interact with each other and aggregate irreversibly as to reduce their surface energy. Solid matrices including noble gas matrices [60], glasses [61], and zeolites [62,63,64] are widely selected for incubating silver nanoclusters with only a few atoms. However, the large size of solid matrices impedes their further application in biomedicine science. The organic templates that encapsulated MNCs were first achieved using water-soluble dendrimer (PAMAM) as the scaffold by Dickson et al. in 2002 [51]. Since then, various synthesis strategies including the template-assisted method and ligand-induced etching have been developed to fabricate organic ligands-capped luminescent MNCs [10,14]. The formation of MNCs with different capping ligands in solution have been accomplished in various ways, as shown in Figure 5 [34]. The synthesis parameters, such as the reduction method of metal ions, the initial ratio of reactants, and the reaction temperature have a profound effect on the generation of luminescent MNCs. 

Prominently, the emission wavelength of the ligand-protected MNCs exhibits a strong ligand-dependent effect. For example, the emission wavelength of DNA oligomer-capped Ag NCs could be varied from the blue to NIR region by playing with the nucleotide sequence [65]. In addition, the photoluminescence intensity is also dependent on the nucleotide sequence. The photoluminescence of DNA-AgNCs can be enhanced 500-fold when placed in proximity to guanine-rich DNA sequences. Based on this discovery, Yeh et al. designed a fluorescent probe to detect specific nucleic acid targets [66]. This fluorescence probe can easily reach high signal-to-background ratios (S/B ratios) (>100) upon target binding, since it does not rely on Forster energy transfer for quenching. Wu et al. systematically studied the ligand’s role in the fluorescence of gold nanoclusters [33]. They found that the photoluminescence intensity was scaled with the electron donation capacity of thiol ligands. The photoluminescence was attributed to the charge transfer from the ligands to the metal nanoparticle core (i.e., LMNCT) through the Au-S bonds. These ligands with electron-rich atoms (e.g., N, O) or groups (e.g., –COOH, NH_2_) can largely promote fluorescence via surface interactions, as shown in Figure 6. Luo et al. demonstrated that even only the aggregated Au(I)−thiolate complexes induced by solvents or cations could generate strong photoluminescence. The formation of aurophilic bonds provided the impetus for aggregation; then, denser and more rigid aggregates were formed. The emission from the aggregates was attributed to ligand-to-metal charge transfer (LMCT) or ligand-to-metal−metal charge transfer (LMMCT) from the sulfur atom in the thiolate ligands to the Au atoms, and subsequent radiative relaxation, which was most likely via a metal-centered triplet state. Pyo et al. reported an efficient strategy to enhance the photoluminescence of gold clusters (Au_22_(SG)_18_) by rigidifying its gold shell with bulky tetraoctylammonium (TOA) cations [67]. The luminous quantum yield could improve up to approximately 60% at room temperature. The luminescence was ascribed to the LMMCT triplet state from the gold shell, which exhibits a strong rigidifying effect. They conclude that the inter-complex aurophilic Au(I)···Au(I) interactions are unlikely in the well-separated dimeric gold shell of the Au_25_(SG)_18_ cluster encapsulated in dendrimer (PAMAM), since these as-synthesized AuNCs were well-separated by the cavity of the dendrimer, and the luminescence can be ascribed to the ligand-to-metal charge transfer (LMCT) effect, rather than the LMMCT effect [26]. Londono-Larrea et al. successfully synthesized water-soluble naked gold nanoclusters (AuNC_naked_) by using only NaOH (the reductant) and HAuCl_4_ [68]. They found that the nonluminescent AuNC_naked_ could be transferred to strong luminescent when passivated with different thiols and adenosine monophosphate. The photoluminescence of the passivated NCs was clearly attributed to the ligand–AuNC surface interaction. 

These results indicated that the surface ligands of MNCs were also pivotal to their photoluminescence properties. Ligand-to-metal charge transfer (LMCT) and ligand-to-metal−metal charge transfer (LMMCT or LMNCT) mechanisms were proposed to understand the principle of the luminescence process. However, some issues are still difficult to understand. For example, these MNCs with identical size could be luminescent or nonluminescent, and they are very sensitive to the delicate change of reaction parameters. 

#### 2.2.2. Metal Valence State Correlated Photoemission

In addition to the cluster size and surface-capping ligands, a surface metal valence state was also a considerable factor for synthesizing highly luminescent MNCs. Zhou et al. observed that the completely different luminescence properties of glutathione protected gold nanoparticles (GS-Au NPs) with identical size but different in metal valence states [44]. The luminescent GS-Au NPs with a core size of approximately 2 nm become nonluminescent after treatment by NaBH_4_, noting that the core size was sustained. If this is true (same size), obviously the metal centered quantum confinement effect (QCE) can not answer this peculiar observation. The authors concluded that the high content of gold(I) (40–50%) in the luminescent nanoparticles verified by X-ray photoelectron spectroscopy (XPS) was responsible for the unique optical properties of the luminescent gold nanoparticles. The luminescence lifetime highly depends on the excitation wavelength; when excited at approximately 420 nm, it emits orange colors with a long lifetime of several microseconds at approximately 565 nm, while, if excited at 530 nm, the same color is emitted with a short lifetime with nanoseconds (Figure 7). This interesting observation of dual PL emissions with different lifetimes indicates the presence of two completely different chromophores. The authors pointed out that the degeneration in energy of the triplet and singlet excited states in the luminescent gold nanoparticles led to the same wavelength emissions due to the change of metal charge valence state. Wu et al. also observed that the fluorescence of Au_25_(SC_2_H_4_Ph)_18_ could be largely enhanced by increasing their oxidation charge state (from −1 up to +2) using oxidants such as O_2_, H_2_O_2_, Ce(SO_4_)_2_, etc. [69,70]. In these reports, all the evidence confirmed the influence of the valence state of a metal ion or core on the PL emissions, and the PL emission of MNCs was originated from the interband or intraband transitions of the outmost shell electrons. However, it is important to note that the increasing of the electropositivity of the metal core, i.e., the high charge state of metals, can also promote the interactions between the surface ligands and metal core, which cannot completely exclude the role of ligand assembly to tune the PL emission.

#### 2.2.3. Self-Assembly Governed Photoemission

The intrinsic defect of luminous MNCs was their limited quantum yield, since these ultra-small clusters with high surface-to-volume ratio easily suffer from the solvent molecules and oxygens, which could quench their photoluminescence [71]. Many MNCs only exhibit photoluminescence under low temperature and are nonluminescent at room temperature, which largely limits their practical application [72]. Aggregation-induced emission (AIE) is an efficient strategy for improving the photoluminescence performance of lumigens. It has attracted tremendous attention since the first observation in organic chromophores by Tang’s group in 2001 [73]. Generally, the aggregation of organic dyes could lead to photoluminescence quenching due to the formation of detrimental species such as excimers [74,75]. In contrast, the luminescence of (luminigens) AIEgens is conspicuously enhanced via aggregation due to the strong restriction of the intramolecular vibrations (RIV) and rotations (RIR) [76]. Recently, this fascinating AIE effect has been regarded as an efficient strategy for optimizing the photoemission performance of nanoclusters by directing their spatial construction [45,56,77,78]. The alternation of nanoarchitectures greatly influences the charge transfer and energy conduction process of MNCs, and thus their photoelectric performance. However, the physical origin of AIE effect is not clearly addressed. In fact, if we carefully checked the structures and PL properties of AIEgens, they showed very similar PL properties with MNCs, such as a very strong solvent effect and large Stock shift. It is possible that their PL origins come from the same physical principle.

Jianping Xie’s group and Erkang Wang’s group first reported that the nonluminescent oligomeric Au(I)−thiolate complexes and weak luminescent CuNCs in aqueous solution could generate strong luminescence upon aggregation induced by weak polar solvents or divalent cations [56,79], and their luminescence enhancement was assigned to the AIE effect. Very recently, Jia et al. demonstrated that the photophysical features of the thiolated AgNCs were dependent on their morphology, which was controlled by the solvents-induced aggregation [41]. The ordered structure of AgNCs assemblies was well defined by XRD and TEM techniques. Subsequently, Benito et al. observed the mechanochromic luminescence properties in copper iodide clusters [46]. Two kinds of crystalline polymorphs with green and yellow emission were obtained. Upon mechanical grinding, the green emissive polymorph exhibits great modification of its emission from green to yellow, as shown in Figure 8. XRD analyses indicated that the crystalline packing was damaged, and an almost complete amorphous state was formed, which implies an assembly architecture-dependent emission property of CuNCs. Obviously, the change of PL properties triggered by simply grounding cannot be simply answered by metal-centered QCE, since individual CuNCs remains unchanged. The authors suggested that the Cu-Cu interactions were responsible for the luminescence properties. Upon mechanical stimulation, destruction of the crystalline structure led to shortening of the Cu−Cu bond in the cluster core and resulting red shift of emission wavelength from green to yellow. These results definitely established the important role of cuprophilic interactions in the mechanochromic mechanism of CuNCs. A similar mechanism was used by Zhang to explain the PL emission of metal nanoclusters (Cu, Au, Ag) self-assemblies with different assembly morphology [45,78,80,81,82,83]. These as-synthesized nonluminescent individual CuNCs could generate strong photoemission when assembled to a crystalline packing structure, similar to the AIE effect, noting that the emission wavelength could be easily adjusted by controlling the assembly structure. As shown in Figure 9, CuNCs exhibit yellow-green emission in nanosheets but blue emission in nanoribbons. A mechanochromic property was also observed in which the blue emissive nanoribbon could transfer to yellow emissive with less crystallinity. 

The relationship between the compactness of assemblies and the emission was summarized as follows. (1) “High compactness reinforces the cuprophilic Cu(I)···Cu(I) interaction of inter- and intra-NCs, and meanwhile, it suppresses intramolecular vibration and rotation of the capping ligand of 1-Dodecanethiol (DT), thus enhancing the emission intensity of Cu NCs. (2) The emission energy depends on the distance of Cu(I)···Cu(I); the improved compactness increases the average Cu(I)···Cu(I) distance by inducing additional inter-NCs cuprophilic interaction, and therewith leads to the blue shift of NCs emission” [45]. Noting that it is counterintuitive for improving the compactness could lead to the increase of Cu(I)···Cu(I) distance. The authors explain this abnormal phenomenon by inducing additional inter-NCs cuprophilic interaction between the neighboring NCs. However, it needs to be reiterated that cuprophilic interactions could only generate when the adjacent Cu···Cu distances are in the range of the van der Walls interaction distance (generally less than 3.6 Å); however, the distance between these two neighboring NCs is at a nanometer scale, which is far beyond the effective distance of metallophilic interactions [32,84,85]. Thus, the rationality of the LMCT and/or LMMCT model was challenged. However, the fact remains that the change of spacing distance between adjacent surface ligands in varied morphologies is definitively confirmed, which further evidenced the paramount role of surface ligand packing to tune the PL properties.

## 3. Our P Band Intermediate State (PBIS) Model Dominates the Photoluminescence Emission of MNCs

Even though the metal-centered free-electron model based on the QCE and, subsequent LMCT and/or LMMCT mechanism, in some sense, explained some important PL emission phenomena, the elucidation and origin of optoelectronic properties is diverse and contradictory, at every point inviting inquiry and debate over a decade. Since 2014, this continuous and long-term study in my group and collaborators has been carried out to understand the nature of the photoluminescence emission of metal nanoclusters and related quantum nanostructures. We first provided the key evidences that the distribution of surface-protecting ligands on the metal core played a paramount role to tune the optoelectronic properties of noble metal NCs. However, the basic chemical principle hidden behind abnormal optical phenomena has been troubling us, such as its room temperature phosphorescence enhancement, surface ligand selectivity, unprecedented large Stokes Shift, tunable optical absorption, newborn electronic band structure, etc. Very recently, using metal NCs as a model system, by judiciously manipulating the delicate surface ligand interactions at the nanoscale interface of a single metal nanocluster, superlattice and mesoporous materials, together with a careful control of the solvophobicity and solvophilicity of the ligands, the resulting interplay of various noncovalent interactions can lead to the precise modulation of optoelectronic properties of metal NCs. A completely new p band intermediate state (PBIS) model was proposed to understand the origin of PL emission of all related quantum nanodots, which completely challenges the metal-centered quantum mechanics for the elucidation of PL emission of metal NCs. We definitively identify that the PBIS stems from the overlapping of p orbitals of the paired or more adjacent heteroatoms (O and S) from the surface-protected ligands on the metal NCs, which can be considered as a dark state at the metal–NCs interface to activate the triplet site of the surface chromophores.

In our earlier investigations, in order to fully understand the structure of luminescent MNCs, we used water-soluble polymers poly(methacrylic acid) (PMAA) as templates to encapsulate AgNCs with small size (2−5 nm) [86]. By precisely designing the experiments’ parameters, we found that the photoluminescence of AgNCs showed high selectivity on the surface-anchoring ligands and strong dependence on the valence state of surface metal. That was, the strong fluorescence of carboxyl-protected AgNCs disappeared when the surface functional ligands were changed to sulfonic acid groups, even though the metal core was sustained, indicating the key role of ligand type on the regulation of PL properties. Based on these experimental results, a core-shell structural model was proposed to understand the nature of photoluminescence of Ag NCs. As shown in Figure 10, the fluorescence from the AgNCs was attributed to ligand-to-metal−metal charge transfer (LMMCT) from Ag(I)-carboxylate complexes (the oxygen atom in the carboxylate ligands to the Ag(I) ions) to the Ag atoms and subsequent radiative relaxation. In this report, we also followed the classical QCE to elucidate the PL emission of MNCs. However, we highlighted the pivotal role of surface ligands to regulate the PL properties of MNCs.

To further clarify the emission origin of MNCs, we separately discussed the functions of surface ligands and metal core to the photoluminescence. Metal-centered emission (MCE) and ligand-centered emissions (LCE) were simultaneously observed in AgNCs [87]. Luminescent water-soluble AgNCs protected by sulfydryl ligands with different functional groups (carboxyl, amino, alkyl) were successfully fabricated using a modified cyclic reduction−decomposition approach. Two distinct photoemissions with emission peaks at approximately 580 and 665 nm were observed, as shown in Figure 11. The emission at approximately 580 nm with a small Stokes shift (30 nm), narrow peak width (full width at half maxima (FWHM) approximately 30 nm), short lifetime (1.6 ns), and low quantum yield (<1%) was ligand-independent, since it could be observed in all of these as-synthesized AgNCs with different capping ligands. In contrast, the emission at approximately 665 nm with a large Stokes shift (>200 nm), broad peak width (FWHM approximately 100 nm), relative longer lifetime (180 ns), and higher quantum yield (approximately 10%) was ligand-dependent, and it could only be observed in AgNCs protected by carboxyl. They were attributed to MCE and LCE, respectively. The MCE was highly dependent on the metal core, and the LCE was highly related to the surface ligands and solvent environment. Accordantly, the emission at approximately 580 nm was pH independent, and the emission at approximately 665 nm exhibits a strong pH-dependent effect (Figure 12a–c). Based on these observations, a new ligand synergistic emission effect was proposed to understand the PL emission of MNCs. That is, the amino-correlated nπ* state provides a pivot to bridge the carboxyl correlated ππ* and nπ* states to enhance the charge transfer efficiency between different surface electronic states. Consequently, the photoluminescence quantum yields were significantly improved (approximately 1% to 10%), as shown in Figure 12. Very recently, our unpublished results showed that the assignment on the emission at approximately 580 nm is probably not right. Thus, the ligand synergistic emission mechanism needs further optimization.

If the metal-centered QCE dominates the PL emission of MNCs, the optical properties should not be significantly affected by the solvent effect. However, it is not the case. Very recently, we unexpectedly observed a strong solvent-induced enhancement effect of carboxyl-protected water-soluble AgNCs, indicating the invalidity of the well-accepted metal centered QCE model [88]. The emission QY of AgNCs could be largely improved from approximately 1% to 40% by a simple solvent-stimulated strategy (Figure 13). The fluorescence-phosphorescence dual solvoluminescence (SL) of water-soluble metal nanoclusters (NCs) at room temperature was observed. The photoluminescence was originated from the self-assembly of surface ligands, which was induced by solvent stimulation, that is, the clustering of surface ligands, as illustrated in Figure 14. The clustering of surface carbonyl groups was promoted in two different ways: (1) the strong interaction between carboxylate ligands and the metal core, and (2) strong n → π* interactions between these two adjacent carbonyl groups [89,90,91]. The clustering of carbonyl groups could lead to the extension of conjugation and efficient delocalization of electrons in overlapped C=O double bonds between neighboring carbonyl groups. In addition, the molecular conformation becomes more rigid, which could largely restrict the intramolecular vibrations (RIV) and rotations (RIR). Finally, the photoluminescence was largely enhanced. We first proposed that the exact chromophore of metal NCs for aggregation-induced emission (AIE) mechanics was originated from the clustering carbonyl groups. This interpretation gives a completely new insight into the luminescent principle of MNCs, but we cannot completely preclude the role of the metal core for the PL emission. 

Very most recently, based on the complimentary characterizations of steady-state absorption, excitation, and time-resolved PL spectroscopy, we definitively identify that the induced p band intermediate state (PBIS) stems from the overlapping of p orbitals of the paired or more adjacent heteroatoms (O and S) from the surface-protected ligands on the metal NCs, which can be considered as a dark state at the metal–NCs interface to activate the triplet site of the surface chromophores by enhancing intersystem crossing [43]. Figure 15 showed that the water-soluble and oil-soluble MNCs with identical size synthesized by a ligand-exchange strategy (denoted as Au NCs@GSH and Au NCs@DT, respectively) exhibited the strong solvent-enhanced PL emission behavior, and that it is very interesting that the number and intensity of PL emission was strongly dependent on the type of used surface-protective ligands, even though the metal core remains intact. The more convincing and powerful evidence for the PBIS model comes from the luminescent mesoporous silica nanoparticles functionalized by nonluminescent organosilanes with amino and carbonyl groups free of any metals, which bears the same spectroscopic properties as metal NCs. These surface-modified MSNs with target organic functions showed a very strong and tunable photoluminescence emission due to the clustering of nonluminescent chromophores in the confined nanospace [92,93,94,95,96,97,98,99]. It is importantly noted that the selectively used organosilane in the functional groups contains heteroatoms with unpaired lone electrons, such as oxygen (O), nitrogen (N), and sulfur (S), as used protecting ligand molecules for the synthesis of metal NCs [100,101,102]. These nonluminescent functional groups assembled on a non-metallic surface generated the same photoluminescence emission with MNCs, as shown in Figure 16. We definitively confirmed that the pairing of surface ligands could serve as an exact chromophore for PL emission, which has nothing to do with the metal core. 

A descriptive ligand-assembly-mediated interfacial p band intermediate state (PBIS) model was proposed to understand the origin of the PL emission of MNCs, as shown in Figure 17. At the nanoscale interface of the metal nanocluster core or in the confined nanomesopores, the adjacent ligands locally interact with each other to form Rydberg matter-like clusters by the overlapping of p-orbitals from O, N, S, and P on proximal carbonyl/thiol groups with high-energy lone-pair electrons [103,104]. The delocalization of the high-energy electrons in coupled p orbitals in ligand-directed molecular architectures produces a new overall lower energy state, the so-called PBIS, which acts as an intermediate (or dark) state to tune PL emissions by intersystem crossing where the energy gap between the singlet excited state and the intermediate p band state governs the direction and the extent of the electron transfer. Time-dependent density functional theory (TD-DFT) calculation indicated that the energy level of the p-band center is very sensitive to the local proximity ligand chromophores at heterogeneous interfaces, which further confirmed the validity of the PBIS model. 

Based on this model, which is shown here as an atypical example, the abnormal polymorph-dependent emission wavelength phenomenon in anisotropic superlattices could be readily understood [45]. As illustrated in Figure 18, in the highly compact structure (nanoribbon), due to the strong intercalation between DT molecules in the layers, the d spacing between two DT molecules is larger than that in nanosheets; then, the overlapping or the interactions between paired or neighboring DT molecules are weakened, which answers that the blue-shift emission of the nanoribbons is due to the declined electron delocalization between adjacent DT molecules. 

The PBIS model was also successful to explain the long-term debated and self-contradictory size-dependent and size-independent PL phenomena of MNCs: the smaller the metal nanoparticle size, the more coordinated unsaturated metal atoms are exposed with increased surface-to-volume rations, resulting in the strong binding and high surface coverage of surface coating ligands, consequently emitting enhanced PL intensity with low energy due to the maximum overlapping of p orbitals from surface ligands, where the MNCs provided an ideal nanoscale interface for the strong binding interactions of adsorbed molecules, for example, the surface-protective ligands. When the nanoparticle size is fixed, the intensity and wavelength of PL are dependent on ligand coverage or densities: a high Au-S coordination number (CN) and high surface coverage results in stronger PL emission at long wavelengths because of the close packing of surface ligands, whereas a low Au-S CN and a low surface coverage make weak PL emissions with high energy. Obviously, the PL emission of metal nanoclusters is strongly dependent on the close packing or self-assembly of targeted surface ligands; thus, the used template for MNCs must contain electron-rich heteroatoms, including oxygen (O), nitrogen (N), sulfur (S), and phosphorus (P). 

## 4. Summary and Outlooks

Numerous and complicated superficial factors, such as the size and type of MNCs, the surface ligands, the valence state of surface metal, the self-assembly, and even the solvent effect affect the luminescent properties of metal nanoclusters, which interferes with the understanding of the luminescent nature of metal nanoclusters. Using metal NCs as a model system, by judiciously manipulating the delicate surface ligand interactions at the nanoscale interface together with a careful calculation of time-dependent density functional theory (TD-DFT), we proposed a completely new p band intermediate state (PBIS) model to elucidate the origin of PL emission of MNCs, which is completely different from the metal-centered quantum confinement mechanics (QCE) for the elucidation of PL emission of metal NCs. This mechanism could be universal to explain the PL emission nature of other quantum dot systems, including carbon and graphene nanodots, transition metal dichalcogenide (TMDC) nanomaterials, luminescent metal organic framework (MOFs), luminescent perovskites, and even organometallic complexes [105,106,107,108,109,110], because the overlapping of p orbitals at the confined nanointerface and nanospace could not be precluded [111]. Even though the PBIS model could qualitatively describe the PL properties, the exact dynamics and kinetics of electron transition of the excited state were not clear. In the near future, ultrafast time-resolved technology such as transient absorption and fluorescence up-conversion at the femtosecond scale will provide some reliable measurement tools to describe the relaxation and decay pathways of excited electrons. Most importantly, with respect to the paramount importance and universality of ligand and interface interaction at the nanoscale interface, the proof of concept of PBIS not only elucidates the fundamental physical principles driving nanoscale PL emission phenomena, but it also provides completely new insights to understand the interface-confined nanocatalysis on the molecule level. The presence of a new interface state with a low energy level could act as an extra channel to promote election transfer, which reduces the energy barrier of redox reaction, similar to the role of the transition state called in the heterogeneous catalysis [112,113]. At the same time, if the atomic orbitals of reactants interact with the PBIS, the binding strength of the reactant with the metal active site could be optimized (the classical Sabatier principle), which will improve the selectivity and lifetime of catalysts [114]. Of course, PBIS could be self-spontaneously formed between two adsorbed neighboring reactants at the heterogeneous nanoscale interface, which will also accelerate the chemical reaction depending the surface coverage. Hence, this significant conceptual advance will be of immediate interest to a broad readership of researchers in the nanocatalysis science.

## Figures and Tables

**Figure 1 nanomaterials-10-00261-f001:**
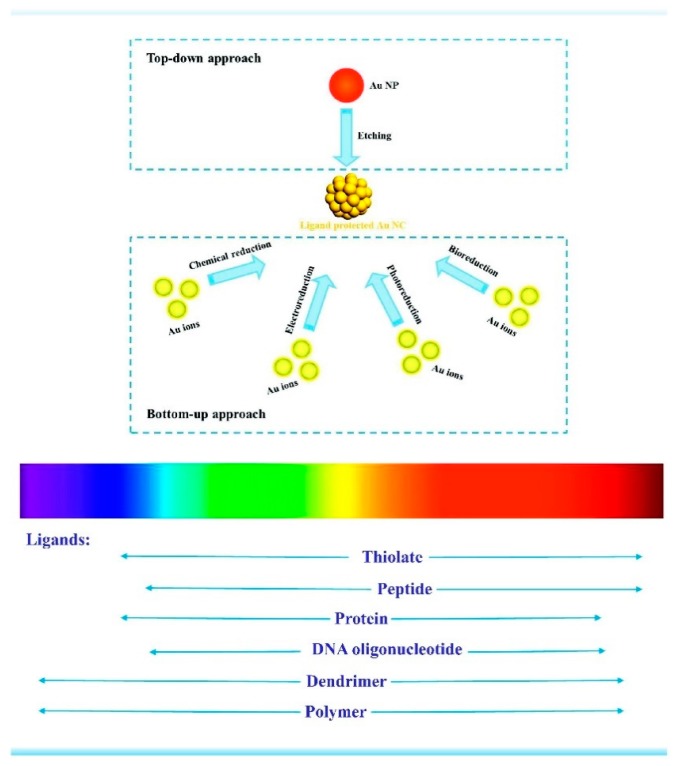
Schematic illustration of synthetic strategies of luminescent AuNCs and effect of ligands on their photoluminescence. Reprinted with permission from Ref. [14]. Copyright (2017) Elsevier.

**Figure 2 nanomaterials-10-00261-f002:**
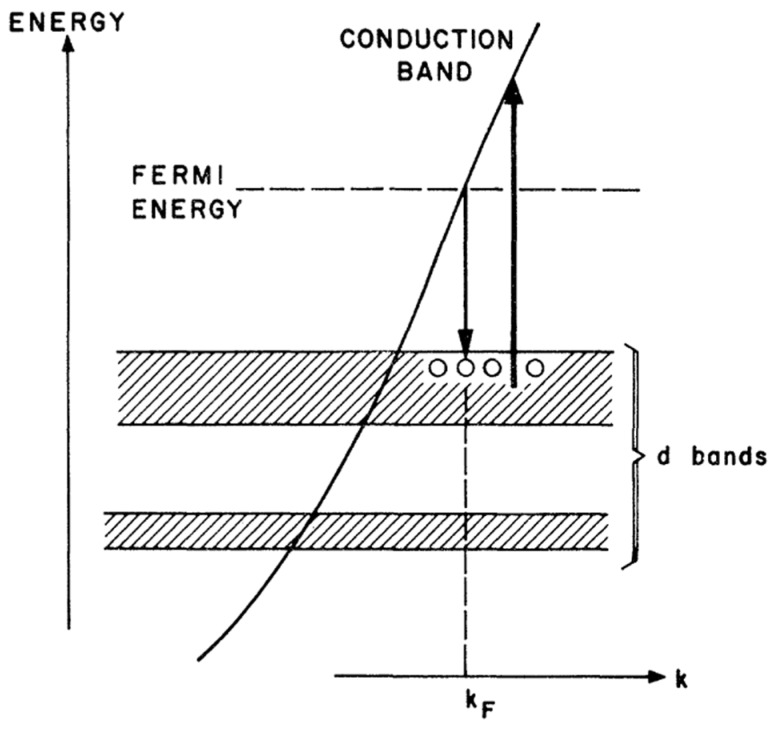
Schematic band structure of a noble metal showing the excitation and recombination transitions. Reprinted with permission from Ref. [29]. Copyright (1969) American Physical Society.

**Figure 3 nanomaterials-10-00261-f003:**
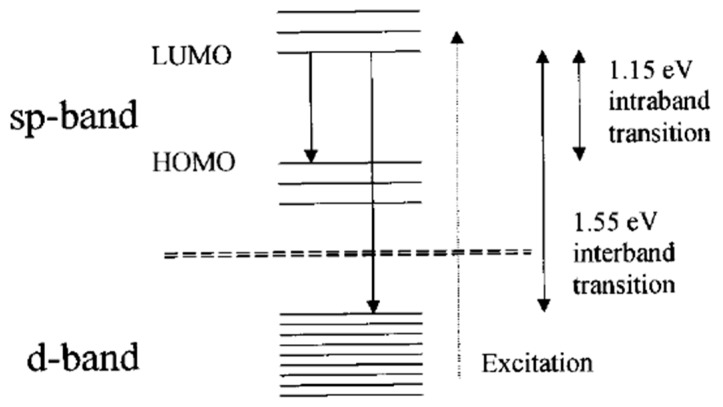
Solid-state model for the origin of the two luminescence bands: The high-energy band is proposed to be due to the radiative interband recombination between the sp and d-bands, while the low-energy band is thought to originate from radiative intraband transitions within the sp-band across the HOMO-LUMO gap (highest occupied molecular orbital and lowest unoccupied molecular orbital). Note that intraband recombination has to involve the prior nonradiative recombination of the hole in the d-band created after excitation with an (unexcited) electron in the sp-band. Reprinted with permission from Ref. [30]. Copyright (2002) American Chemical Society.

**Figure 4 nanomaterials-10-00261-f004:**
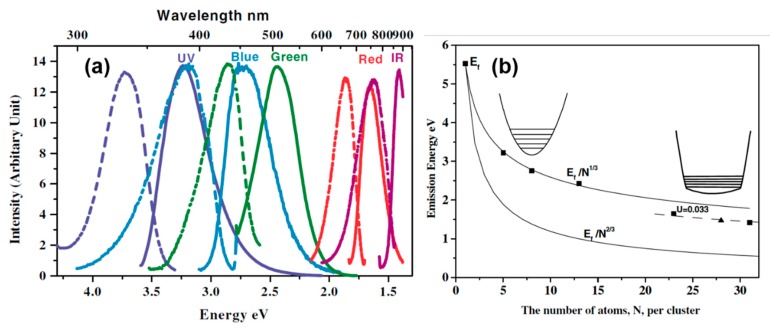
(**a**) Excitation (dashed) and emission (solid) spectra of different sizes of PAMAM-Au NCs. Excitation and emission maxima shift to longer wavelengths with increasing initial Au concentration, suggesting that increasing the nanocluster size leads to lower energy emission. (**b**) Correlation of the number of Au atoms per cluster (N) with the photoemission energy of Au NCs. Reprinted with permission from Ref. [26]. Copyright (2004) American Physical Society.

**Figure 5 nanomaterials-10-00261-f005:**
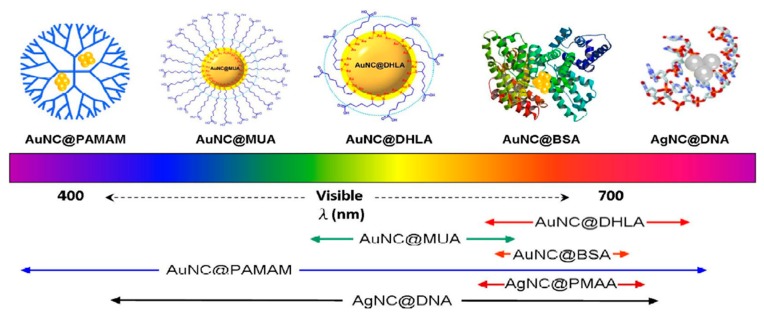
Representative luminescent noble–metal nanoclusters scaled as a function of their emission wavelength superimposed over the spectrum. Protected molecules show different capabilities to tune the emission wavelength of metallic nanoclusters from current reports.

**Figure 6 nanomaterials-10-00261-f006:**
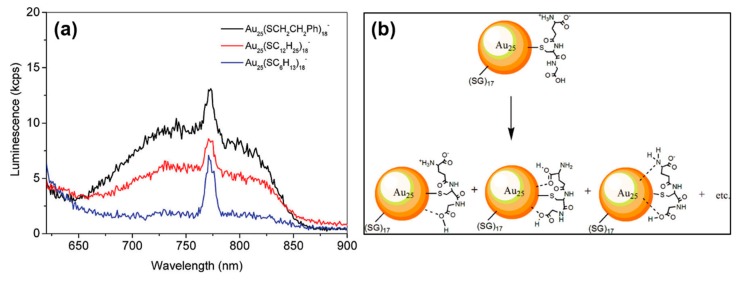
(**a**) Weak fluorescence of [Au_25_(SR)_18_]- with different R groups (–C_2_H_4_Ph, –C_12_H_25_, and –C_6_H_13_). (**b**) Possible Interactions of amine and carboxyl groups of glutathione (–SG) ligands to the gold surface. Reprinted with permission from Ref. [33]. Copyright (2010) American Chemical Society.

**Figure 7 nanomaterials-10-00261-f007:**
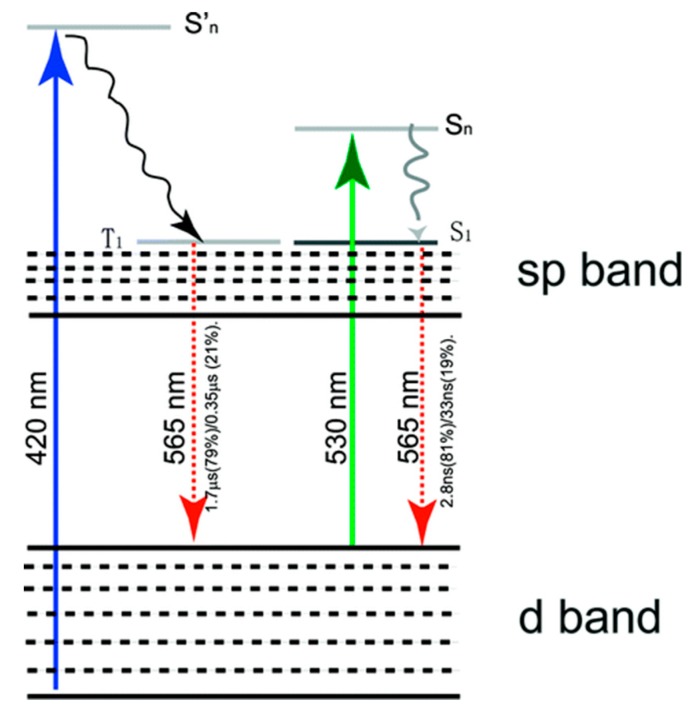
A possible optical transition scheme for orange-emitting GS-AuNPs where the luminescence originates from transitions between d and sp bands. When the NPs are excited at 420 nm, the electrons will be relaxed from triplet states in sp bands to some ground states in d bands, leading to microsecond emission. Once the excitation wavelength is shifted to 530 nm, the electrons will be decayed from singlet excited states in the sp band to singlet states in ground states and give out a nanosecond emission. The triplet and singlet excited states in the luminescent gold nanoparticles are degenerate in energy. Reprinted with permission from Ref. [44]. Copyright (2010) American Chemical Society.

**Figure 8 nanomaterials-10-00261-f008:**
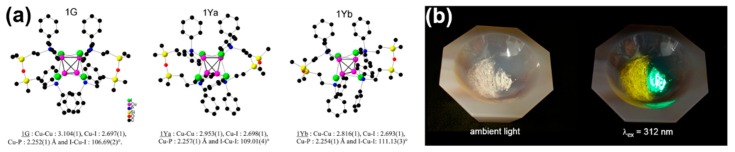
(**a**) Molecular structure of clusters 1G and 1Y and the mean of selected bond lengths. Hydrogen atoms have been omitted for clarity. (**b**) Photos of the ground (left) and intact (right) crystalline powder of 1G under ambient light and under UV irradiation at 312 nm (UV lamp) at room temperature. Reprinted with permission from Ref. [46]. Copyright (2014) American Chemical Society.

**Figure 9 nanomaterials-10-00261-f009:**
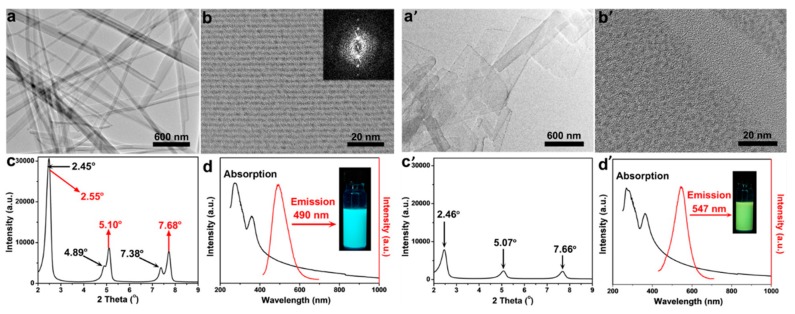
(**a**) TEM image of the ribbons from Cu NCs’ self-assembly. (**b**) HRTEM image of the ribbons. Inset: the Fourier transform image. (**c**) Small-angle region of XRD pattern. (**d**) Steady-state absorption (black) and emission (red) spectra of the ribbons in chloroform. Inset: the fluorescent image with 365 nm excitation. (**a’**) TEM image of the sheets from Cu NCs’ self-assembly. (**b’**) HRTEM image of the sheets. (**c’**) Small-angle region of XRD pattern. (**d’**) Steady-state absorption (black) and emission (red) spectra of the sheets in chloroform. Inset: the fluorescent image with 365 nm excitation. Reprinted with permission from Ref. [45]. Copyright (2015) American Chemical Society.

**Figure 10 nanomaterials-10-00261-f010:**
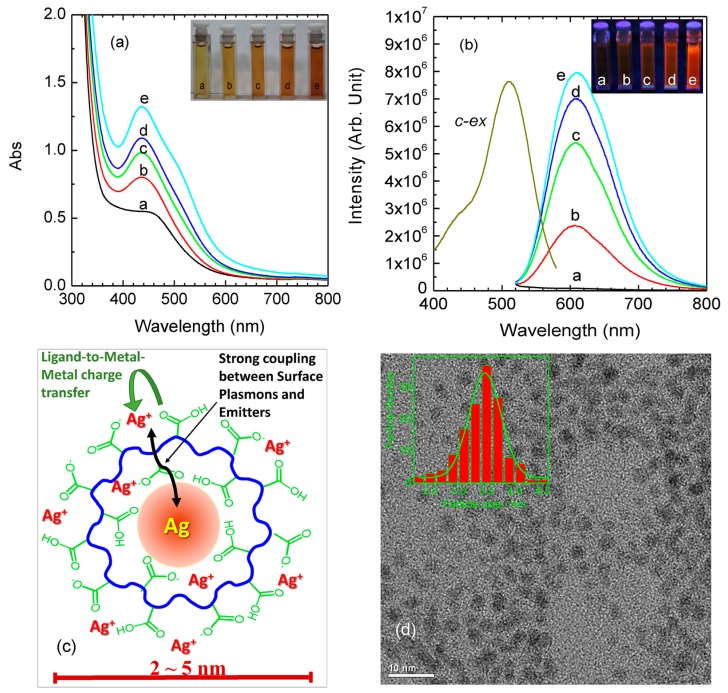
(**a**) UV−vis absorption of the freshly prepared Ag-carboxylate NCs using PMAA as scaffold at various radiation times. (**b**) PL spectra of the Ag-carboxylate NCs. Inset images show the corresponding photographs of Ag-carboxylate NCs under room and UV light exposure at λ = 365 nm. (**c**) Schematic illustration of the structures of our luminescent Ag-carboxylate with Ag^+^-carboxylate complexes shell. (**d**) TEM image of freshly prepared Ag-carboxylate NCs (histogram describes the statistical distribution of the particle size, which is approximately 3.5 nm). Reprinted with permission from Ref. [86]. Copyright (2014) American Chemical Society.

**Figure 11 nanomaterials-10-00261-f011:**
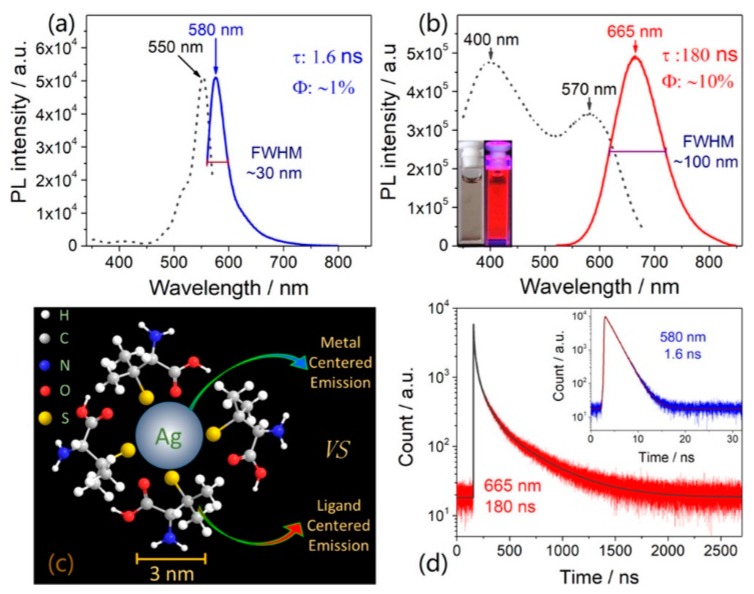
Excitation and emission spectra of the narrow approximately 580 nm emission (**a**) and broad approximately 665 nm emission (**b**). Inset images show the corresponding photographs of D-penicillamine-capped silver nanoclusters (DPA-AgNCs) under room and UV light exposure at λ=365 nm. (**c**) Schematic illustration of the metal-centered and ligand-centered emission mechanisms of DPA-capped AgNCs. (**d**) Time-resolved luminescence decay profiles of DPA-AgNCs in water solution measured at 665 and 580 nm (inset), excited at 400 and 550 nm, respectively. Reprinted with permission from Ref. [87]. Copyright (2019) American Chemical Society.

**Figure 12 nanomaterials-10-00261-f012:**
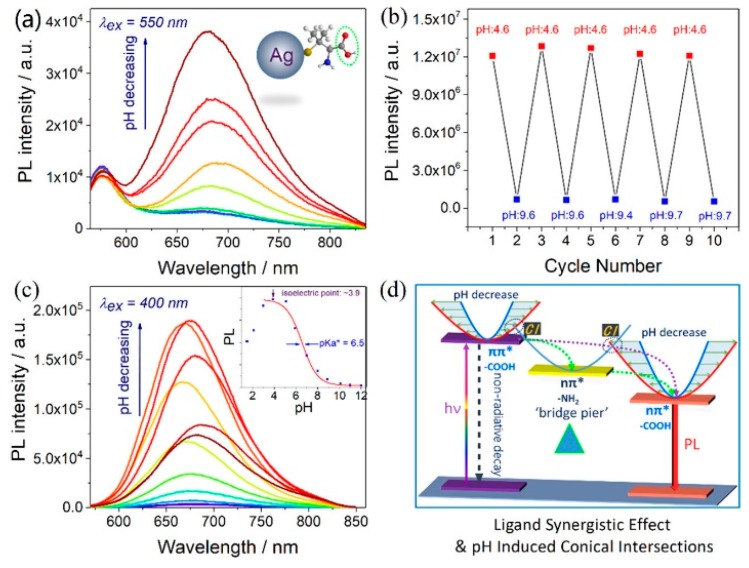
(**a**) Photoluminescence spectra of as-synthesized DPA-AgNCs in different pH conditions. (**b**) PL intensity of DPA-AgNCs upon cyclic switching of the pH between 4.6 and 9.7. (**c**) PL spectra and PL intensity (inset) of DPA-AgNCs with changing pH values. PL spectra were excited at 400 nm; pH value over the range 1.5−12. (**d**) Schematic illustration of the ligand synergistic effect and pH-induced conical intersections. Reprinted with permission from Ref. [87]. Copyright (2019) American Chemical Society.

**Figure 13 nanomaterials-10-00261-f013:**
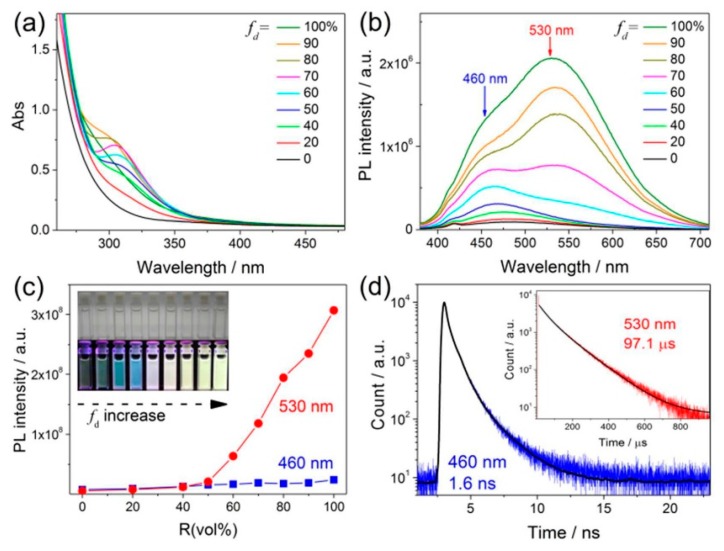
(**a**) UV−vis absorption and (**b**) photoluminescence spectra of polymethyl vinyl ether-alt-maleic acid-capped silver nanoclusters (PMVEM-Ag NCs) in the varied volume fraction *f*_d_ of DMSO in the mixed solvent (*f*_d_ = V_DMSO_/V_DMSO+water_). Photoluminescence was excited at 365 nm. (**c**) Correlations of the emission intensities of the two peaks centered at approximately 460 and approximately 530 nm versus *f*_d_. (Inset) Photographs of PMVEM-Ag NCs at different *f*_d_ under visible (top row) and UV (bottom row) light. (**d**) Time-resolved luminescence decay profiles of PMVEM-Ag NCs in DMSO measured at 460 and 530 nm (inset), respectively. Reprinted with permission from Ref. [88]. Copyright (2017) American Chemical Society.

**Figure 14 nanomaterials-10-00261-f014:**
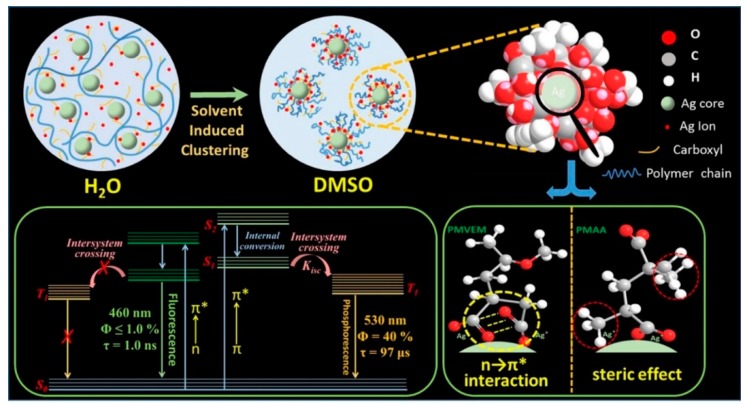
Schematic illustration of the solvent-induced clustering process of Ag NCs (top) depending on the unique molecular structure of the polymer used as a template for the synthesis of Ag NCs (bottom-right) and the energy-level structure of Ag NCs in water solution and DMSO solution (bottom-left). Reprinted with permission from Ref. [88]. Copyright (2017) American Chemical Society.

**Figure 15 nanomaterials-10-00261-f015:**
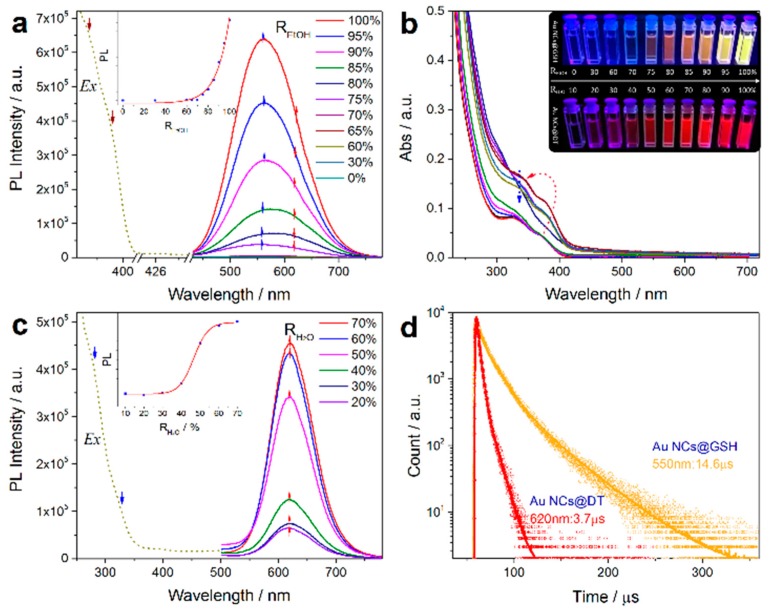
Solvent-induced ligand-dependent optical absorption and emissions. Photoemission and excitation spectra of water-soluble Au NCs@GSH (**a**) and oil-soluble Au NCs@DT (**c**) in mixed solvents with different volume fractions of R (Inset, relationship between the luminescence intensity and R (R_EtOH_ = Vol_EtOH_/Vol_EtOH + H2O_, R_H2O_ = Vol_H2O_/Vol_EtOH + H2O_), the spectra were recorded at 0.5 h after the sample preparation). (**b**) Ultraviolet-visible (UV−vis) absorption spectra of Au NCs@GSH in mixed solvents with different R_EtOH_. Inset shows the digital photos of water-soluble Au NCs@GSH and oil-soluble Au NCs@DT in mixed solvents of ethanol and water with varied volume fractions of R_EtOH_ and R_H2O_ under UV light. (**d**) Time-resolved luminescence decay profiles of solvent-induced luminescent Au NCs@GSH and Au NCs@DT. Reprinted with permission from Ref. [43]. Copyright (2019) Springer Nature. GSH: glutathione.

**Figure 16 nanomaterials-10-00261-f016:**
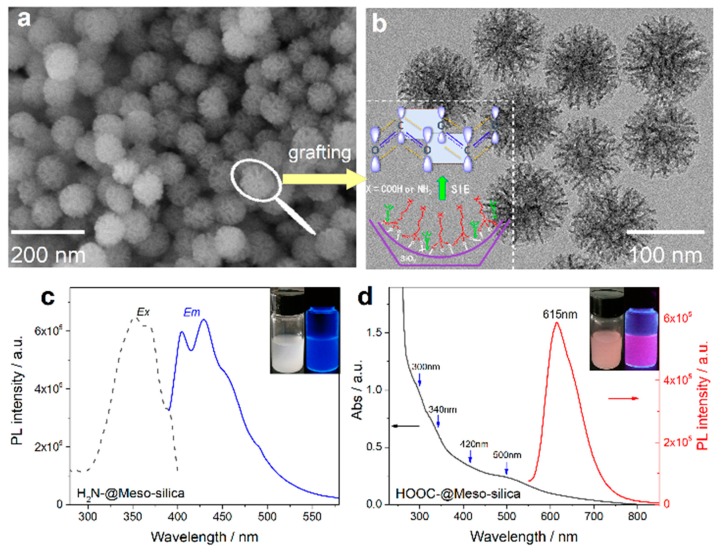
Ligand assembly in the mesoporous silica nanoparticles (MSNs) free of metals and their tunable luminescent properties. Scanning electron microscopy (SEM) (**a**) and TEM (**b**) images of as-synthesized fluorescent mesoporous silica nanoparticles. The inset shows the assembly of amino and carbonyl groups in the confined nanopores. (**c**) Excitation and emission spectra of aminopropyl-functionalized MSNs. (**d**) Absorption and emission spectra of propylsuccinic-functionalized MSNs. Reprinted with permission from Ref. [43]. Copyright (2019) Springer Nature.

**Figure 17 nanomaterials-10-00261-f017:**
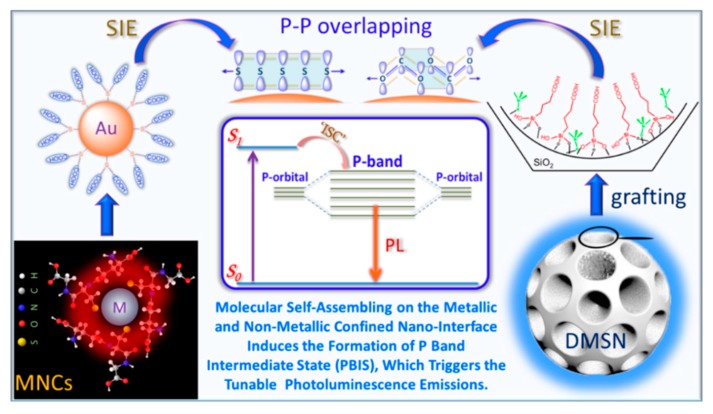
Ligand-assembly-mediated p band intermediate state (PBIS) dominates photoluminescence emission. Schematic illustration of the ligand exchange process and solvent-induced emission (SIE) properties of Au NCs (inset: the energy-level structure of Au NCs in water and ethanol mixed solution). The p band formed by the overlapping of p orbitals of electron-rich sulfur and oxygen heteroatoms of well-organized surface ligands is used as an intermediate state or dark state to tune the optoelectronic properties. Please see more details on the pioneering conceptual PBIS model. Reprinted with permission from Ref. [43]. Copyright (2019) Springer Nature.

**Figure 18 nanomaterials-10-00261-f018:**
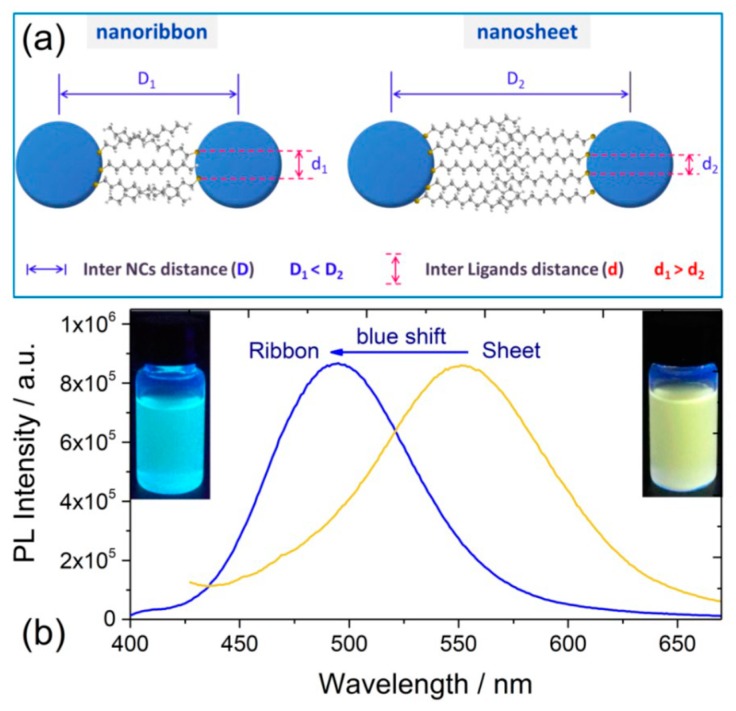
(**a**) Schematic structures of a Cu superlattice with different nanoribbon and nanosheet morphology. (**b**) Corresponding emission spectra of Cu NCs with nanoribbon and nanosheet morphology. The real distance between two thiol groups in the nanoribbon Cu superlattice is larger than that in the nanosheet Cu superlattice, indicating the loose packing of DT molecules on the metal core. The emission peak exhibits a blue shift from 550 nm to 490 nm when the morphology of the Cu superlattice changed from a nanosheet to nanoribbon, which implies a change of the packing model of DT molecules. Reprinted with permission from Ref. [43]. Copyright (2019) Springer Nature.

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
