# Peer review of "Origin of the Photoluminescence of Metal Nanoclusters: From Metal-Centered Emission to Ligand-Centered Emission"

_nanomaterials, 2020, doi:10.3390/nano10020261_

Round 1

Reviewer 1 Report

The paper presents an interesting review on the origin of the photoluminescence of Metal Nanoclusters. I think that it deserves to be published. I found no issues in it and therefore it can be published as it is.

Author Response

Thank the referee's very positive comments on our manuscript.

Reviewer 2 Report

Zhang and coworkers review the current understanding of the origin of photoluminescence from metal nanoclusters. Historical accounts are given which describe observations and explanations of size-dependent emission followed by size-independent emission, the latter of which focuses on ligands, metal valences, and aggregation. Finally, the authors review their p band intermediate state theory to tie together disparate descriptions on the origin of photoluminescence from metal nanoclusters. Overall, the organization of the review is easy to follow. The differentiator of this review to others (Yu et al. Coord. Chem. Rev. 2019, 378, 595–617; Kand and Zhu, Chem. Soc. Rev. 2019, 48, 2422) written on the same topic and implicitly covering a lot of the same material is the inclusion of the p band intermediate state theory. Several minor revisions are suggested as listed below to improve the manuscript in order to be suitable for publication.

General comments:

The competing reviews (Yu et al. Chem. Rev. 2019, 378, 595–617; Kang and Zhu, Chem. Soc. Rev. 2019, 48, 2422) should be cited Acronyms are not used consistently or even defined at the proper places The quality of the English is low which makes the content delivery difficult To improve readability, it is suggested that short summaries are given at the end of each section of the manuscript Fluorescence and photoluminescence are used interchangeably. Because many different systems and postulations are being discussed, and to improve readership, it is recommended that photoluminescence be used as a general term unless specific timescales are being discussed.

Specific comments:

Page 1, line 33: capitalize Stokes Page 1, line 36: The sentence as written is incorrect. Photoemission from quantum dots is well-known to be tunable based on their size and surface ligands. Page 2, line 51: LMMCT is not defined. In addition to defining the acronym, it is recommended that the authors clearly explain the difference between LMCT and LMMCT Page 3, lines 93-95: Citations are required to support this statement. It has been shown that plasmon resonance can enhance photoluminescence from small metal nanoparticles (Cai et al. ACS Nano 2018, 12, 976-985) Page 4, line 118: remove “extraordinary”. This is editorializing and is not descriptive of the effect Page 4, line 131: “bond” should be “band” Page 5, line 149: “GSH” not defined Page 7, line 226: “GS” not defined Figure 8A is difficult to read. Its size should be increased.

Author Response

General comments:

The competing reviews (Yu et al. Chem. Rev. 2019, 378, 595–617; Kang and Zhu, Chem. Soc. Rev. 2019, 48, 2422) should be cited Acronyms are not used consistently or even defined at the proper places The quality of the English is low which makes the content delivery difficult To improve readability, it is suggested that short summaries are given at the end of each section of the manuscript Fluorescence and photoluminescence are used interchangeably. Because many different systems and postulations are being discussed, and to improve readership, it is recommended that photoluminescence be used as a general term unless specific timescales are being discussed.

Answer: This is a really nice comment for the review paper, especially for the reader's understanding. as suggested by referee, we gave a short summary for each part at the end of the separated sections, and in the main text, now photoluminescence (PL) was used as a general term. In addition, the two references suggested by refree was added,which are really related to our topic.

Page 1, line 33: capitalize Stokes Page 1, line 36: The sentence as written is incorrect. Photoemission from quantum dots is well-known to be tunable based on their size and surface ligands. Page 2, line 51: LMMCT is not defined. In addition to defining the acronym, it is recommended that the authors clearly explain the difference between LMCT and LMMCT Page 3, lines 93-95: Citations are required to support this statement. It has been shown that plasmon resonance can enhance photoluminescence from small metal nanoparticles (Cai et al. ACS Nano2018, 12, 976-985) Page 4, line 118: remove “extraordinary”. This is editorializing and is not descriptive of the effect Page 4, line 131: “bond” should be “band” Page 5, line 149: “GSH” not defined Page 7, line 226: “GS” not defined Figure 8A is difficult to read. Its size should be increased. 

Answer: Done as suggested.

In addition, the plasmon resonance enhanced photoluminescence is very hot topic, but beyond this review range, since the plasmon resonance NPs generally exhibits a large particle size (>3.0)and in priciple quench the PL emission. However, the reference ( ACS Nano2018, 12, 976-985) provided by referee showed the opposite PL behavior. This is a very interesting paper, which needs further invesigations.

Finally, because the manuscript was greatly rewitten, we strongly suggested that referee reread the revised edition. Thanks again the referee's constructive suggestions and comments.

Reviewer 3 Report

The paper is a review on the evolution of luminescent mechanism models of MNS, starting with pure metal-centered quantum confinement mechanics to
ligand centered p band intermediate state (PBIS) model via a transitional ligand to metal charge transfer (LMCT or LMMCT) mechanism.

The review is bringing a comprehensive view within this field of research, the information are described clearly in an organized manner, with referenced work. 

There is one aspect that could be taken into consideration, the authors should consider including a section with the applications of photoluminescent metal nanoclusters.

Author Response

Thanks the referee's very positive comment on our review paper. 

THe refree is right that, in this review paper, we did not discuss on the recent advances on the applications of metal NCs since in this invited theme issue there are two review papers focusing on the applications of MCNs. anyway, as suggested by referee, several review papers on the applications, especially for the catalysis, were added in the follwing:

Yu et al. Coord. Chem. Rev. 2019, 378, 595–617; Kand and Zhu, Chem. Soc. Rev. 2019, 48, 2422.